# INVERSE ENGINEERING DIFFUSION:
# DERIVING VARIANCE SCHEDULES WITH RATIONALE

## ABSTRACT

A fundamental aspect of diffusion models is the variance schedule, which governs the evolution of variance throughout the diffusion process. Despite numerous studies exploring variance schedules, little effort has been made to understand the variance distributions implied by sampling from these schedules and how it benefits both training and data generation. We introduce a novel perspective on score-based diffusion models, bridging the gap between the variance schedule and its underlying variance distribution. Specifically, we propose the notion of sampling variance according to a probabilistic rationale, which induces a density. Our approach views the inverse of the variance schedule as a cumulative distribution function (CDF) and its first derivative as a probability density function (PDF) of the variance distribution. This formulation not only offers a unified view of variance schedules but also allows for the direct engineering of a variance schedule from the probabilistic rationale of its inverse function. Additionally, our framework is not limited to CDFs with closed-form inverse solutions, enabling the exploration of variance schedules that are unattainable through conventional methods. We present the tools required to obtain a diverse array of novel variance schedules tailored to specific rationales, such as separability metrics or prior beliefs. These schedules may exhibit varied dynamics, ranging from rapid convergence towards zero to prolonged periods in high-variance regions. Through comprehensive empirical evaluation, we demonstrate the efficacy of enhancing the performance of diffusion models with schedules distinct from those encountered during training. We provide a principled and unified approach to variance schedules in diffusion models, revealing the relationship between variance schedules and their underlying probabilistic rationales, which yields notable improvements in image generation performance, as measured by FID.

Changes to the initial submission have been highlighted in orange.

## 1 INTRODUCTION

Diffusion models have emerged as powerful tools in the realms of image, audio, and video generation, facilitating remarkable progress in capturing complex data distributions. The general framework of diffusion models was first introduced in Sohl-Dickstein et al. (2015) and later re-popularized by Ho et al. (2020), both using a discrete time Markov chain to transform the data distribution to noise. Song et al. (2020) relaxed the framework to operate in continuous time by rephrasing the distribution transforming process as a Markov process following a stochastic differential equation (SDE). This approach allows the deployment of a variety of SDE and ordinary differential equation (ODE) solvers for the reverse process, leading to significant performance gains (Karras et al., 2022). An essential component of diffusion models is the definition of a variance schedule, which describes the evolution of the variance in the underlying stochastic process over time. Despite numerous studies on different variance schedules, little effort has been made to characterize the distribution of variance and justify specific schedules beyond empirical performance observations.

To foster a better understanding of the variance schedule in score-based diffusion models, we interpret the inverse function of the variance schedule as a cumulative distribution function (CDF), with an associated probability density function (PDF) induced by a probabilistic rationale that weights

the importance of variances throughout the diffusion process. Our rephrased framework has several theoretical and practical contributions:

**Introducing rationale:** We propose the notion of a rationale that enables designing variance schedules from a distribution of variances, ranging from rapid convergence to zero variance to prolonged periods in high-variance regions, or smooth transition from high to low variance.

**Inverse engineering diffusion:** We show that a driftless forward process is fully determined by the choice of a rationale enabling us to "inverse engineer" the diffusion process by selecting a rationale and constructing the variance schedule from the inverse function of the respective CDF.

**Choosing any probability density:** We demonstrate that a variance schedule can be defined for any rationale and its corresponding probability density. Even if the resulting CDF lacks a closed-form solution, the generalized inverse of the CDF can be used to derive the variance schedule.

**Simplifying design choices:** We show that loss weighting simplifies to choosing a different rationale for training than for sampling. Furthermore, we explore the effect of switching variance schedules with different rationales after training.

Our experimental results confirm the effectiveness of our approach, demonstrating that variance schedules derived from a rationale, can lead to improved image quality. We highlight one rationale in particular, which would have been unattainable through conventional methods.

## 2 BACKGROUND

We begin with a brief overview of the continuous-time formulation of score-based diffusion models through the lens of SDEs. Song et al. (2020) propose modeling the distribution transformation process from data to noise as a stochastic process in continuous time, following the forward dynamics

$$d\mathbf{x} = f(t)\mathbf{x}dt + g(t)d\mathbf{w}_t, \tag{1}$$

where $\mathbf{w}_t$ is a standard Wiener process with a continuous function $f : [0, T] \to \mathbb{R}$ and continuous diffusion function $g : [0, T] \to \mathbb{R}$. Remarkably, this formulation provides an exact reverse-time process with dynamics (Anderson, 1982)

$$d\mathbf{x} = \left[ f(t)\mathbf{x} - g(t)^2 \nabla_{\mathbf{x}} \log p_t(\mathbf{x}) \right] dt + g(t) d\mathbf{w}_t, \tag{2}$$

where $\nabla_{\mathbf{x}} \log p_t$ is the unknown score function associated with the marginal density $p_t$ of $\mathbf{x}$ at time $t \in [0, T]$. The unknown score function is approximated using a parameterized score model $\mathbf{s}_\theta$, which is trained via score-matching Song et al. (2020).

To efficiently sample from the forward process at any time, we can adopt a probabilistic perspective. Conditioning the forward process on its starting value $x(0)$ yields the closed-form transition kernel (Song et al., 2020; Karras et al., 2022)

$$p_{0t}\left(x(t)|x(0)\right) = \mathcal{N}\left(x(t);\ s(t)x(0), s(t)^2\sigma(t)^2\mathbf{I}\right), \tag{3}$$

where transitions, related to the function $f$ are described by $s(t)$ and the dynamic relation of the function $f$ and the diffusion function $g$ is incorporated into the variance schedule $\sigma(t)$. See appendix A for the detailed formulae.

## 3 METHOD

Following recent advances in driftless variance schedules (Song et al., 2020; Karras et al., 2022; Song et al., 2023; Karras et al., 2023) we focus here on forward processes without mean shifts, with $s(t) \equiv 1$, and refer to Appendix A for the general case. For a driftless forward process, the transition kernel simplifies to $p_{0t}\left(x(t)|x(0)\right) = \mathcal{N}\left(x(t); x(0), \sigma(t)^2\mathbf{I}\right)$ such that the conditional

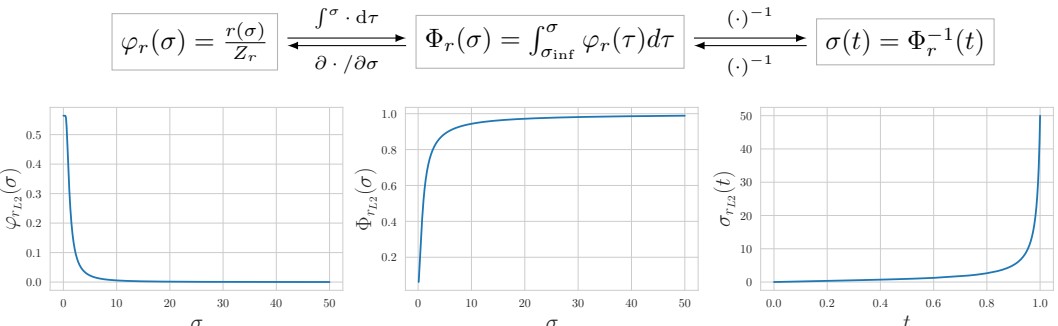

Figure 1: The relation of the normalized probabilistic rationale (i.e., PDF), its CDF and its corresponding variance schedule, exemplified for the squared L2-norm rationale (see section 3.3.2).

marginals are fully characterized by the variance schedule $\sigma : [0, T] \to \mathbb{R}$ and the diffusion function can be rephrased in terms of the variance schedule $\sigma(t)$ and its derivative via (Karras et al., 2023)

$$\sqrt{\int_0^t g(u)^2 \mathrm{d}u} = \sigma(t) \Leftrightarrow g(t) = \sqrt{2\sigma(t)\frac{\partial \sigma(t)}{\partial t}}. \tag{4}$$

In what follows, we exploit the above equivalence to define the forward dynamics of the distribution transforming process based on a rationale that weights the importance of different variance regimes.

## 3.1 A UNIFIED PERSPECTIVE

In order to better understand the characteristics of sampling variance proportional to uniformly distributed time-steps in a variance schedule, we firstly introduce the notion of a rationale.

**Definition 3.1** *Given an interval $\mathcal{I} = (\sigma_{\inf}, \sigma_{\sup}) \subset \mathbb{R}$, we call the positive function $r : \mathcal{I} \to \mathbb{R}_{>0}$ a rationale on $\mathcal{I}$ if $r$ is integrable with a finite normalizing constant $Z_r := \int_{\mathcal{I}} r(\tau)\mathrm{d}\tau < \infty$.*

The rationale $r(\sigma(t))$ scores the benefit of sampling a particular value $\sigma(t)$ at any point in time $t$ of the forward diffusion process. By definition, any such rationale can be normalized to obtain a PDF on the interval $\mathcal{I}$. The idea is to add a meaning (i.e., rationale) to oversampling certain variances and then sample proportionally. For all rationales, a variance schedule exists such that sampling from the rationale is equivalent to sampling from the respective variance schedule with uniformly distributed inputs, as will be explored in the following. Such rationales can incorporate metrics or prior knowledge on the diffusion process, allowing for variance schedules in training and inference that follow a well-understood PDF.

To sample from a rationale, a few steps have to be completed. It is common to define a variance schedule $\sigma(t)$ on the interval $t \in [0, 1]$ (Song et al., 2020; Karras et al., 2022). The inverse function of the variance schedule can then be interpreted as a CDF. Applying the Smirnov transform, sampling from the variance schedule with uniformly distributed inputs is equivalent to sampling proportionally to the PDF of the respective CDF. The connection between the PDF and the respective variance schedule will be explained in the following.

Given a rationale $r$ defined on $\mathcal{I}$ with corresponding normalizing constant $Z_r$ we can construct a PDF $\varphi_r$ on $\mathcal{I}$ by

$$\varphi_r(\tau) := \frac{r(\tau)}{Z_r}, \quad \tau \in \mathcal{I} \tag{5}$$

with corresponding CDF

$$\Phi_r(\sigma) := \int_{\sigma_{\inf}}^{\sigma} \varphi_r(\tau)\mathrm{d}\tau. \tag{6}$$

Finally, we define the variance schedule $\sigma_r$ induced by the rationale $\varphi_r$ via the inverse function

$$\sigma_r(t) := \Phi_r^{-1}(t) \tag{7}$$

of the CDF $\Phi_r$. Following the Smirnov transform, we are able to sample from $\varphi_r$ utilizing the inverse of the CDF (i.e., the variance schedule) with uniformly distributed time-steps, which is equivalent to sampling proportionally to the PDF

$$\sigma \underset{t \sim \mathcal{U}[0,1]}{\sim} \sigma_r(t) \Leftrightarrow \sigma \sim \varphi_r. \tag{8}$$

Thus, we implicitly retrieve a variance schedule $\Phi_r^{-1} = \sigma(t)$ that is based on the rationale $r$. Figure 1 visualizes the relation between the variance schedule and its underlying rationale. The rationale sufficiently defines the variance schedule and by the Smirnov transform, every variance schedule has a rationale. It is important to note that under the Smirnov transform, $\Phi^{-1}$ can be the generalized inverse of the CDF

$$\sigma_r(t) = \Phi_r^{-1}(t) = \inf\{\tau : \Phi_r(\tau) \geq t\}, \tag{9}$$

for all $t \in [0, 1]$. Hence, we do not rely on a closed-form solution of the inverse function to obtain the variance schedule. However, a closed-form solution for $\sigma(t)$ is preferable, as equation 9 comes at additional computational cost (see algorithm 1), which can be precomputed. This allows us to reason about the effectiveness of variance schedules in training and inference beyond empirical observation and directly allocate more compute on sections in the diffusion process where $r$ deems the density arising from $\sigma$ to be of significance.

## 3.2 Rephrasing diffusion with rationale

For a driftless forward process, the variance schedule only depends on the diffusion function $g$ that can be rephrased in terms of the variance schedule $\sigma(t)$ and its derivative $\partial\sigma(t)/\partial t$ according to Equation (4). Recent works have shown promising results using driftless diffusion processes (Song et al., 2020; Karras et al., 2022; Song et al., 2023; Karras et al., 2023), motivated by diffusion being the dominant operation and possibly yielding better results for SDE and ODE solvers when the drift of the forward process is omitted (Karras et al., 2022).

Now that we have explored the relation between the variance schedule and its inverse function (the CDF) in equation 7, as well as the respective PDF in equation 5, we can rephrase the forward- and reverse diffusion processes in terms of the rationale $r$. First, we note that by the inverse function rule and the generalized inverse of the CDF (see equation 9), we can obtain the first derivative of $\sigma_r$ without the explicit closed-form solution of $\sigma_r$ itself:

$$\frac{\partial\sigma_r(t)}{\partial t} = \frac{\partial\Phi_r^{-1}(t)}{\partial t} = \frac{1}{\frac{\partial\Phi_r}{\partial t}\left(\Phi_r^{-1}(t)\right)} = \frac{1}{\varphi_r\left(\Phi_r^{-1}(t)\right)}. \tag{10}$$

Following eqs. (4), (7) and (10), this allows us to rephrase the diffusion function $g$ as:

$$g(t) = \sqrt{\frac{2\Phi_r^{-1}(t)}{\varphi_r\left(\Phi_r^{-1}(t)\right)}}. \tag{11}$$

Here, $g$ is an expression of the variance schedule $\sigma_r$ (i.e., $\Phi_r^{-1}$) and the PDF $\varphi_r$. Thus, using equation 4, we may further rephrase the SDE of the forward diffusion process in equation 1 as:

$$d\mathbf{x} = f(t)\mathbf{x}dt + \sqrt{\frac{2\Phi_r^{-1}(t)}{\varphi_r\left(\Phi_r^{-1}(t)\right)}}d\mathbf{w}_t, \tag{12}$$

and the reverse process in equation 2 as:

$$d\mathbf{x} = \left[f(t)\mathbf{x} - \left(\sqrt{\frac{2\Phi_r^{-1}(t)}{\varphi_r\left(\Phi_r^{-1}(t)\right)}}\right)^2 \nabla_{\mathbf{x}}\log p_t(\mathbf{x})\right]dt + \sqrt{\frac{2\Phi_r^{-1}(t)}{\varphi_r\left(\Phi_r^{-1}(t)\right)}}d\mathbf{w}_t. \tag{13}$$

Since both $\varphi_r$ and $\sigma_r$ are induced by the rationale $r$, the SDEs in eqs. (12) and (13) are entirely governed by the choice of $r$.

Prior works reason about the variance schedule inducing the diffusion process (Karras et al., 2022). With equation 11, we take it a step deeper, highlighting what rationale a variance schedule is following by virtue of $r$. The connection between the rationale and the inverse function of the CDF, that is the variance schedule, allows us to "inverse engineer" the diffusion function $g$ by choosing a rationale.

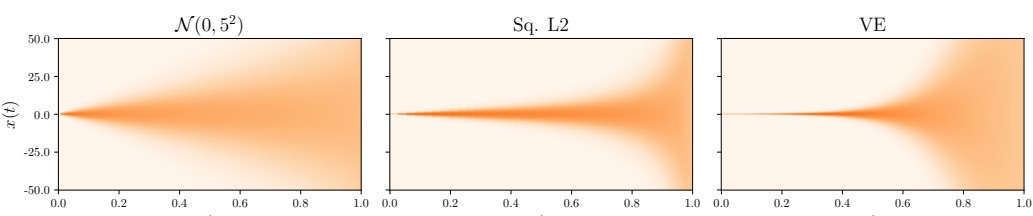

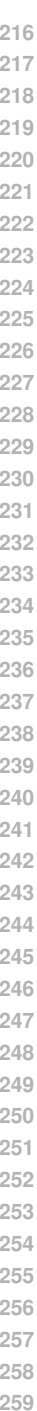

Figure 2: Visualization of the density $p_{0t}(x(t)|\mathbf{0})$ of a forward diffusion process without drift, for variance schedules of the respective rationales $\mathcal{N}\left(0, 5^2\right)$, squared L2-norm, and VE. All plots have a shared y-axis and the timesteps of the diffusion process have been scaled to the interval $(0, 1]$. A deeper Orange implies higher density, White implies low density.

### 3.3 CHOOSING A RATIONALE

As shown in section 3.2, driftless forward diffusion processes are defined by a rationale. Choosing a good rationale is crucial to obtaining a diffusion process that benefits training or sampling. In this work, we condition the score function $\mathbf{s}_\theta$ directly on the standard deviation $\sigma_r(t)$ at time $t$, rather than $t$ itself, such that models can be used with different variance schedules in sampling. In the following, we will explore rationales of established variance schedules, as well as introduce a novel variance schedule that is based on a separability metric. The impact of all rationales on the density $p_{0t}$ of the forward diffusion process is visualized in fig. 2, we observe different dynamics, ranging from rapid convergence towards zero variance to prolonged periods in high-variance regions. Additional rationales that are not discussed further can be found in appendix C.

#### 3.3.1 INVERSE OF ESTABLISHED VARIANCE SCHEDULES

We can explore the inverse function of state-of-the-art variance schedules that have been hand-crafted. The variance schedule we will use as a reference point for a well-defined schedule is the variance exploding (VE) schedule, as introduced in (Song et al., 2020). With $t \in [0, 1]$ and $\sigma \in [\sigma_{\min}, \sigma_{\max}]$, we can derive

$$\Phi_{\text{VE}}^{-1}(t) = \sigma_{\min}\left(\frac{\sigma_{\max}}{\sigma_{\min}}\right)^t, \tag{14}$$

$$\Phi_{\text{VE}}(\sigma) = \frac{\log\left(\frac{\sigma}{\sigma_{\min}}\right)}{\log\left(\frac{\sigma_{\max}}{\sigma_{\min}}\right)}, \tag{15}$$

$$\varphi_{\text{VE}}(\sigma) = \frac{\partial}{\partial \sigma}\Phi_{\text{VE}}(\sigma) = \frac{1}{\sigma \log\left(\frac{\sigma_{\max}}{\sigma_{\min}}\right)}, \tag{16}$$

where we abbreviate $\sigma(t)$ with $\sigma$. Assuming that the data distribution has zero mean and unit variance, we observe that the PDF (i.e., normalized rationale) $\varphi_{\text{VE}}$ is proportional to the square root of the signal-to-noise ratio (SNR). The rationale favors oversampling variance that is close to zero while striking a balance between high and low variances, but skipping intermediate variances through rapid convergence.

In a recent work by Karras et al. (2022), another approach to training diffusion models by leveraging a different variance schedule in combination with loss weighting and tunable hyperparameters for the variance schedule was published. We provide the underlying rationale of the proposed variance schedule of Karras et al. (2022) in appendix C.4, but will focus on VE for all following experiments, as no hyperparameter selection is required.

#### 3.3.2 SQUARED L2-NORM

As discovered in section 3.3.1, the VE schedule is based on a scaled version of the SNR. We can also explore other metrics such as the mean squared error or squared L2-norm between the max-

imum and minimum values $(v_{\max}, v_{\min})$ of the dataset. This rationale is of particular interest for driftless processes. With this approach, we can ascribe high density to points in time of the diffusion process where distributions of different originating means (i.e., $x(0)$) should yield high separability. Oversampling these regions could lead to more detailed samples. We can define the rationale of the squared L2-norm as follows

$$r_{L2}(\sigma(t)) = \int_{-\infty}^{\infty} |p_{0t}(\mathbf{x}|v_{\max}) - p_{0t}(\mathbf{x}|v_{\min})|^2 \, \mathrm{d}\mathbf{x}. \qquad (17)$$

Applying eqs. (5) and (6), with $v_{\min} = -v_{\max}$, we obtain the CDF

$$\Phi_{r_{L2}}(\sigma) = \frac{1}{Z}\sqrt{2}\left(\mathrm{erf}\left(\frac{v_{\max}}{\sigma}\right) v_{\max}\sqrt{\pi} + \sigma \exp\left(-\frac{v_{\max}^2}{\sigma^2}\right) - v_{\max}\sqrt{2}\right), \qquad (18)$$

which does not have a trivial closed-form solution for its inverse function. However, we can still use the inverse function via eq. (9) and algorithm 1. See appendix C.1 for detailed derivations.

### 3.3.3 GAUSSIAN

Other than metrics, we can also apply prior beliefs, such as the belief that sampling low variance is more important than sampling high variance to generate details. This rationale is of particular interest for a parameterized variance schedule that can be tuned to different data-domains. The normal distribution is commonly used to model probabilistic processes due to its good generalization properties in most use-cases. We can model a Gaussian $\mathcal{N}\left(0, \sigma_{\mathcal{N}}^2\right)$ with zero-mean and variance $\sigma_{\mathcal{N}}^2$ from the rationale

$$r_{\mathcal{N}}(\sigma) = \exp\left(-0.5\frac{\sigma^2}{\sigma_{\mathcal{N}}^2}\right), \qquad (19)$$

Where we apply eqs. (5) and (6) to obtain the CDF

$$\Phi_{r_{\mathcal{N}}}(\sigma) = \mathrm{erf}\left(\frac{\sigma}{\sqrt{2}\sigma_{\mathcal{N}}}\right), \qquad (20)$$

and its inverse, the variance schedule

$$\Phi_{r_{\mathcal{N}}}^{-1}(t) = \mathrm{erf}^{-1}(t)\sqrt{2}\sigma_{\mathcal{N}}. \qquad (21)$$

See appendix C.2 for detailed derivations.

### 3.4 LOSS WEIGHTING

Using the generalized perspective of score-based diffusion models presented in this work, we can show that during optimization, loss weighting is equivalent to altering the rationale and, consequently, the variance schedule. Conventionally, we want to minimize the loss

$$\mathbb{E}_{\sigma\sim\varphi_r, \varepsilon\sim\mathcal{N}(0,\mathbf{I})}\left[\lambda(\sigma)||\mathbf{s}_\theta\left(x + \varepsilon\sigma, \sigma\right) - \varepsilon||^2\right], \qquad (22)$$

where $\lambda(\sigma)$ is a positive weighting function used to emphasize certain noise levels post-hoc after sampling from $\varphi_r$. In light of the equivalence equation 8, this weighting can be understood as a reweighing of the underlying density $\varphi_r$ of the variance schedule $\sigma_r$. This can be achieved by the altered rationale $r_\lambda(\sigma) = \lambda(\sigma)\varphi_r(\sigma)$ and normalizing constant $Z_{r_\lambda} = \int r_\lambda(\tau)\mathrm{d}\tau$, such that the new variance schedule $\sigma_{r_\lambda}$ is defined by

$$\varphi_{r_\lambda}(\sigma) = \frac{r_\lambda(\sigma)}{Z_{r_\lambda}}. \qquad (23)$$

This yields the analogous optimization target

$$\mathbb{E}_{\sigma\sim\varphi_{r_\lambda}, \varepsilon\sim\mathcal{N}(0,\mathbf{I})}\left[||\mathbf{s}_\theta\left(x + \varepsilon\sigma, \sigma\right) - \varepsilon||^2\right], \qquad (24)$$

which samples from $\varphi_{r_\lambda}$ rather than weighting samples from $\varphi_r$ with $\lambda$. See proof in appendix B. This shows that loss weighting alters the underlying PDF and rationale used during optimization but can ultimately be expressed by a different rationale and respective PDF. Adopting this formulation has the added benefit of stable gradients for all $\lambda$, as the weighting happens implicitly via sampling rather than directly multiplying with an arbitrary weight.

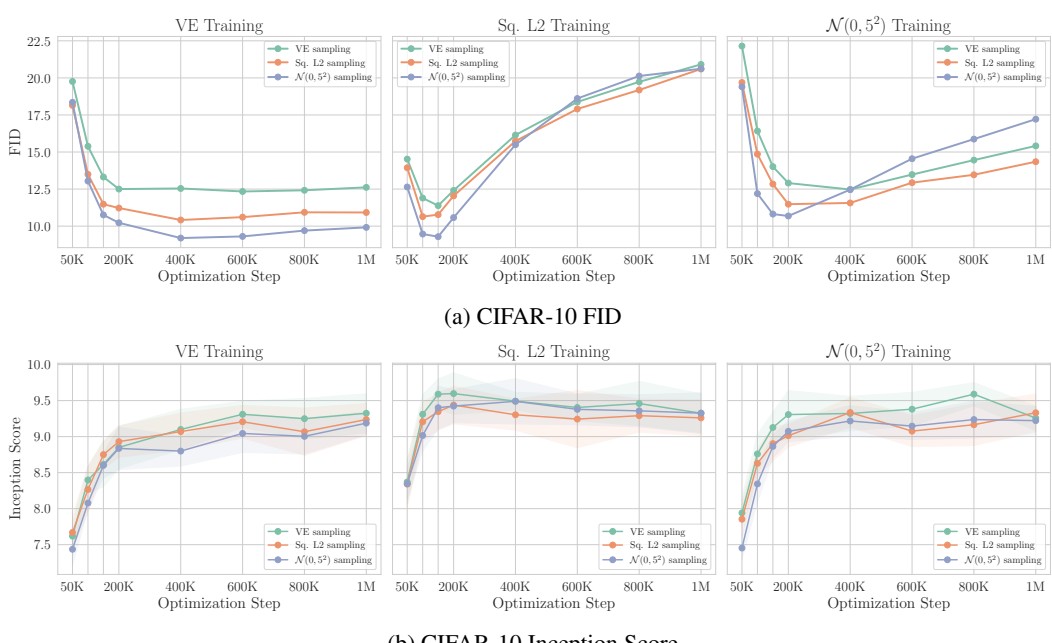

(a) CIFAR-10 FID

(b) CIFAR-10 Inception Score

Figure 3: Metrics throughout entire trainings (class-conditional) on CIFAR-10, all metrics are based on 10K samples for all rationales (VE, squared L2-norm, and $\mathcal{N}(0, 5^2)$). Displayed are FID scores (lower is better) on the CIFAR-10 test-split in (a) and Inception Score (higher is better) with 10K samples in (b). All plots have a shared y-axis. In (b) the standard deviation of the Inception Score is underlayed. Each subplot represents using a specific rationale during training with different rationales used during sampling.

## 4 EXPERIMENTS

We investigated the effects of different rationales on CIFAR-10 (Krizhevsky et al., 2009) and ImageNet-32, a version of ImageNet (Deng et al., 2009) that has been downsampled to $32 \times 32$ images. We explore differences in performance regarding the Frechét-Inception Distance (FID) (Heusel et al., 2017) and the Inception Score (IS) (Salimans et al., 2016) for the variance schedules of respective rationales: VE, squared $L2$-norm, and $\mathcal{N}(0, 5^2)$. We chose $\mathcal{N}(0, 5^2)$, as a representative of variance schedules with gaussian rationale through an ablation study (see appendix D.1 for more details).

Not only do we explore the effect of training with specific variance schedules, but also the effects of sampling with variance schedules that are distinct from the variance schedule used during training. We can achieve this by conditioning the diffusion model on the standard deviation $\sigma_r(t)$ at time $t$ of the forward diffusion process, rather than $t$ itself. This way, we do not only evaluate the effect of variance schedules on the convergence during training, but also the effect on the reverse diffusion process during sampling. While it may appear counter-intuitive to use a different variance schedule during sampling, we note that loss weighting effectively induces a distinct variance schedule during training (see section 3.4) and is commonly featured in the standard framework of score-based diffusion models (Song et al., 2020; Karras et al., 2022). Furthermore, we will show that switching the variance schedule of the diffusion process during sampling can lead to better results. The experiments aim to disentangle what type of rationale (or variance schedule) should be followed when training a diffusion model and when sampling from it.

### 4.1 RESULTS

We ran a total of 1M training steps at a batch size of 128 for all variance schedules when training on CIFAR-10 and a batch size of 512 when training on ImageNet-32. All models were logged every 50K optimization steps. In figs. 3 and 4, we observe the evolution of scores throughout the training.

Table 1: Results for CIFAR-10 (class-conditional). All models were trained from scratch. The table consists of the best FIDs and Inceptions Scores (IS) recorded during training. All metrics were calculated for 10K samples and FIDs with the CIFAR-10 test-split (10K samples). Scores marked with [*] were within the margins of standard deviations respective to the best score in repeated evaluations.

| Training \ Sampling | FID ↓ | | | IS ↑ | | |
|---|---|---|---|---|---|---|
| | **VE** | **L2** | $\mathcal{N}(0, 5^2)$ | **VE** | **L2** | $\mathcal{N}(0, 5^2)$ |
| **VE** | 12.34 | 10.41 | **9.20** | $9.32^* {\pm 0.27}$ | $9.24^* {\pm 0.22}$ | $9.19^* {\pm 0.16}$ |
| **L2** | 11.38 | 10.64 | $9.29^*$ | $\mathbf{9.60} {\pm \mathbf{0.29}}$ | $9.44^* {\pm 0.26}$ | $9.49^* {\pm 0.31}$ |
| $\mathcal{N}(0, 5^2)$ | 12.47 | 11.48 | 10.69 | $9.59^* {\pm 0.16}$ | $9.33^* {\pm 0.20}$ | $9.24^* {\pm 0.26}$ |

All results in figs. 3 and 4 and tables 1 and 2 were computed using the Euler-Maruyama method (Cohen & Elliott, 2015) with 1000 discretization steps. Note that 10K samples were generated to calculate scores for the CIFAR-10 dataset and 50K samples were generated to calculate scores for the ImageNet-32 dataset. Both scores (FID and IS) improve with an increasing number of samples, which is why the scores for CIFAR-10 may appear worse. To calculate the FID we chose the CIFAR-10 test-split (10K samples) and the ImageNet-32 validation-split (50K samples).

We notice that training with the squared $L2$-norm variance schedule results in faster convergence for both CIFAR-10 and ImageNet-32. The model starts overfitting on CIFAR-10 when training with the squared L2-norm and $\mathcal{N}(0, 5^2)$ rationale after around 150K to 200K optimization steps, whereas training with the VE variance schedule results in slower but smoother convergence. The difference in how fast the models converge is clearer to see when training on the larger ImageNet-32 dataset (see fig. 4), where no clear overfitting was observed after 1M optimization steps.

Comparing the novel rationales of the squared L2-norm and $\mathcal{N}(0, 5^2)$ to the VE schedule during sampling, we notice that sampling with the squared L2-norm rationale consistently improves FID scores on CIFAR-10 and is slightly better than VE on ImageNet-32. The $\mathcal{N}(0, 5^2)$ rationale exhibits more varied performance, where it performs better than other schedules when the model has been trained with the squared L2-norm. Comparing the resulting variance schedules w.r.t. their induced density $p_{0t}$ in fig. 2, it may be the case that when sampling from the $\mathcal{N}(0, 5^2)$ rationale, the diffusion models performance is more dependent on intermediate variance, rather than low or high variances. The results in tables 1 and 2 suggest that it is often favorable to switch to a different variance schedule during sampling. These results may appear peculiar and counter-intuitive at first glance. One would assume that it is best to sample with the same variance schedule that was used during training. However, we find that given the Euler-Maruyama method, not all discretizations of the reverse process are equally suitable, given a specific objective for the resulting samples. The inception score is less effected by sampling from different variance schedules than the FID score, as can be observed in all experiments where differences are commonly within overlapping standard deviations of the respective inception scores (see figs. 3 and 4 and tables 1 and 2). A noticeable difference that can be observed with the inception score on both datasets is faster convergence when using the squared L2-norm rationale during training. This is consistent with results that were observed when measuring performance via FID.

In tables 1 and 2, we list the FID and inception scores for all combinations of schedules used during training and sampling. We argue that the training and sampling can be viewed as distinct entities. Training carries out a finite amount of optimization steps, and sampling carries out a finite amount of sampling steps. Both yield a discretization of the underlying continuous dynamic. It is not guaranteed that a discretization that yields good convergence during training also yields optimal samples given the Euler-Maruyama method. Our findings show that the squared L2-norm yields fast convergence and can be paired with the $\mathcal{N}(0, 5^2)$ rationale during sampling to improve results w.r.t. the FID.

## 4.2 IMPLEMENTATION DETAILS

We kept all experiments fair, with an identical U-Net architecture and training strategy for all schedules and datasets. The RAdam (Liu et al., 2019; Kingma & Ba, 2014) optimizer and a constant learning rate of $2 \cdot 10^{-4}$, as proposed by Song et al. (2020), was used for all trainings. The objective

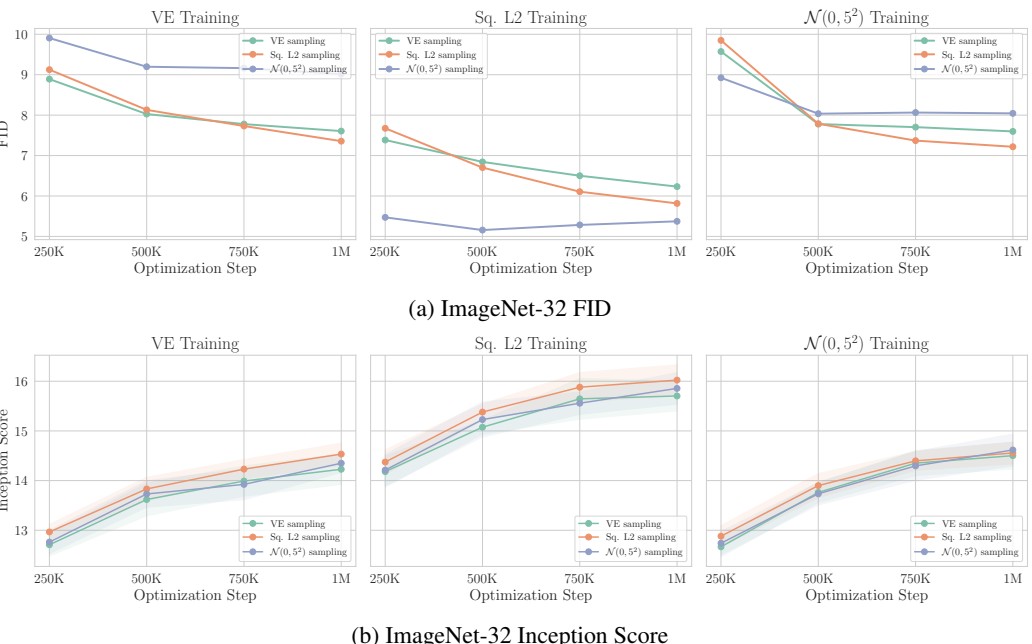

(a) ImageNet-32 FID

(b) ImageNet-32 Inception Score

Figure 4: Metrics throughout entire trainings (class-conditional) on ImageNet-32, all metrics are based on 50K samples for all rationales (VE, squared L2-norm, and $\mathcal{N}(0, 5^2)$). Displayed are FID scores (lower is better) on the ImageNet-32 test-split in (a) and Inception Score (higher is better) with 50K samples in (b). All plots have a shared y-axis. In (b) the standard deviation of the Inception Score is underlayed. Each subplot represents using a specific rationale during training with different rationales used during sampling.

Table 2: Results for ImageNet-32 (class-conditional). All models were trained from scratch. The table consists of the best FIDs and Inceptions Scores (IS) recorded during training. All metrics were calculated for 50K samples and FIDs with the ImageNet-32 validation-split (50K samples). Scores marked with * were within the margins of standard deviations respective to the best score in repeated evaluations.

| Training \ Sampling | FID ↓ | | | IS ↑ | | |
|---|---|---|---|---|---|---|
| | VE | L2 | $\mathcal{N}(0, 5^2)$ | VE | L2 | $\mathcal{N}(0, 5^2)$ |
| VE | 7.61 | 7.36 | 9.02 | 14.23 ±0.31 | 14.53 ±0.22 | 14.35 ±0.18 |
| L2 | 6.23 | 5.82 | **5.16** | 15.70* ±0.30 | **16.02** ±**0.31** | 15.86* ±0.32 |
| $\mathcal{N}(0, 5^2)$ | 7.60 | 7.22 | 8.04 | 14.50 ±0.27 | 14.56 ±0.22 | 14.62 ±0.32 |

of the experiments was not to tweak a specific rationale to achieve state-of-the-art performance (e.g., utilizing predictor-corrector sampling and shifting $\sigma_{\min}$ post-hoc as in Song et al. (2020)), but rather comparing the baseline performance of rationales relative to another to achieve fair and conclusive observations. We use exponential moving averages (EMA) with an EMA-rate of 0.999 for all experiments, as proposed in Song et al. (2020) for the VE schedule. Our U-Net has a base feature size of 128 features, and we pooled three times multiplying the base feature size at each pooled level by 2. Attention (Vaswani et al., 2017) was used on feature maps in the bottleneck, similar to Song et al. (2020). We condition the model on the standard deviation with a sinusoidal embedding, allowing us to choose a variance schedule that is different from the one we trained with during sampling. We scaled the standard deviation $\sigma(t)$ by $0.25 \cdot \log \sigma(t)$ before embedding it for the neural network, as proposed in Karras et al. (2022). When sampling from the diffusion model, we used the Euler-Maruyama method with 1000 steps for all evaluations. Mixed precision with Bfloat16 was used for all experiments. All CIFAR-10 trainings were completed within two and a half A100 GPU days and all ImageNet-32 trainings were completed within ten A100 GPU days; generating 10K samples took approximately 1 hour on a single A100 GPU.

## 5 RELATED WORK

**Modeling of diffusion processes**  The foundational research conducted by Song et al. (2020) presents a cohesive framework for modeling the transformation of distributions through stochastic processes in continuous time, incorporating an exact reverse-time model. Subsequent extensive investigations have explored (Karras et al., 2022; Chen et al., 2023) and expanded upon (Jing et al., 2022; Kim et al., 2022; Huang et al., 2022; Lou & Ermon, 2023; Song et al., 2023; Yoon et al., 2023; Bartosh et al., 2024) the continuous-time perspective on generative models using SDEs.

**Advancements in score-based diffusion models**  While (Karras et al., 2022) offers a comprehensive analysis of the components within the score-based diffusion framework, it does not delve into the underlying density of variance induced by variance schedules. Unlike Song et al. (2020), which conditions the score model during training on the time step via a time embedding, Karras et al. (2022) and Song et al. (2023) directly utilize the marginal variance of the diffusion process for conditioning. However, they do not provide further characterization of the marginal variance distribution in terms of a probability density function and the resulting rationale. In Raya & Ambrogioni (2024), the authors explore the effects of sampling at lower variances for the variance preserving (VP) variance schedule with drift, motivated by spontaneous symmetry breaking.

## 6 CONCLUSION

In this work we introduce a novel perspective on the score-based diffusion framework, which allows to reason about the rationale and effect of sampling variance during the training and inference of score-based diffusion models. Our work highlights the nuanced impact of variance schedules on training and sampling processes for generative models applied to CIFAR-10 and ImageNet-32. By evaluating models with different training and sampling configurations, we observed distinct advantages for each approach. Notably, the squared L2-norm variance schedule demonstrated faster convergence during training but a propensity for overfitting on CIFAR-10. In contrast, the VE variance schedule exhibited slower yet smoother convergence on both datasets.

Remarkably, the best FID scores were achieved when sampling according to the $\mathcal{N}\left(0, 5^2\right)$ rationale, which did not yielded neither fast nor stable convergence during training. This underscores the importance of viewing training and sampling as separate entities. Our findings challenge the intuitive approach of viewing variance schedules to serve the same purpose for both training and sampling, suggesting that different discretization processes may yield better outcomes depending on the specific objectives and solvers.

Futhernmore we proved that loss weighting induces a different variance schedule during training and that using a variance schedule during sampling that is distinct to that, which was used during training can be beneficial. Using the perspective of rationales, we can not only disentangle the implied variance schedule when training with loss weighting, but also better understand the distribution of variance for any given variance schedule via its inverse function.

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

## A  GENERAL FRAMEWORK

In the following, we will outline the general framework of score-based diffusion models. In Song et al. (2020) the authors propose considering a diffusion process in continuous time. Here, the forward process $\mathbf{x}$ solves the SDE

$$\mathrm{d}\mathbf{x} = f(t)\mathbf{x}\mathrm{d}t + g(t)\mathrm{d}\mathbf{w}_t, \tag{25}$$

where $\mathbf{w}_t$ is a standard Wiener process with drift function $f$ and diffusion function $g$. This formulation results in the exact reverse process (Anderson, 1982)

$$\mathrm{d}\mathbf{x} = \left[ f(t)\mathbf{x} - g(t)^2 \nabla_{\mathbf{x}} \log p_t(\mathbf{x}) \right] \mathrm{d}t + g(t)\,\mathrm{d}\mathbf{w}_t, \tag{26}$$

where $\nabla_{\mathbf{x}} \log p_t(\mathbf{x})$ is the unknown score function associated with the marginal density $p_t$ of $\mathbf{x}$. A parameterized score-model $\mathbf{s}_\theta$ can be learned to approximate $\nabla_{\mathbf{x}} \log p_t(\mathbf{x})$.

To efficiently sample from the forward process, we can adopt a probabilistic perspective. Conditioning the forward process on its starting value yields the closed-form transition kernel (Song et al., 2020; Karras et al., 2022)

$$p_{0t}\left(x(t)|x(0)\right) = \mathcal{N}\left(x(t);\ s(t)x(0), s(t)^2\sigma(t)^2\mathbf{I}\right), \tag{27}$$

where transitions, related to the drift function $f$ are described by

$$s(t) = \exp\left(\int_0^t f(u)\,\mathrm{d}u\right) \tag{28}$$

and a variance schedule $\sigma(t)$ is induced by the diffusion function $g$, as well as $s$ via

$$\sigma(t) = \sqrt{\int_0^t \frac{g(u)^2}{s(u)^2}du}. \tag{29}$$

It is evident that these perspectives are interchangeable. The dynamic relation of the drift function $f$ and the diffusion function $g$ is incorporated into the variance schedule $\sigma(t)$. Following Karras et al. (2022), the forward diffusion process can be rephrased by the equivalence

$$\sqrt{\int_0^t \frac{g(u)^2}{s(u)^2}\mathrm{d}u} = \sigma(t) \Leftrightarrow g(t) = s(t)\sqrt{2\sigma(t)\frac{\partial\sigma(t)}{\partial t}}. \tag{30}$$

This finding allows to rephrase the forward process in eq. (1) as

$$\mathrm{d}\mathbf{x} = f(t)\mathbf{x}\,\mathrm{d}t + s(t)\sqrt{2\sigma(t)\frac{\partial\sigma(t)}{\partial t}}\,\mathrm{d}\mathbf{w}_t, \tag{31}$$

and the reverse process in eq. (2) as

$$\mathrm{d}\mathbf{x} = \left[ f(t)\mathbf{x} - \left(s(t)\sqrt{2\sigma(t)\frac{\partial\sigma(t)}{\partial t}}\right)^2 \nabla_{\mathbf{x}} \log p_t(\mathbf{x}) \right] \mathrm{d}t + s(t)\sqrt{2\sigma(t)\frac{\partial\sigma(t)}{\partial t}}\,\mathrm{d}\mathbf{w}_t. \tag{32}$$

In the case of a driftless diffusion process, we have $s(t) \equiv 1$ such that the variance schedule only depends on $g$. In such a case, the diffusion process is characterized by the variance schedule $\sigma$ and its first derivative $\partial\sigma(t)/\partial t$ according to eq. (4). Recent works have shown promising results using driftless diffusion processes (Song et al., 2020; Karras et al., 2022; Song et al., 2023; Karras et al., 2023), motivated by diffusion being the dominant operation and possibly yielding better results for SDE and ODE solvers when the drift of the forward process is omitted (Karras et al., 2022).

## B  LOSS WEIGHTING

Given a weighting function $\lambda$ and a rationale $r$, we can construct a new rationale via

$$r_\lambda(\sigma) = \lambda(\sigma)\varphi_r(\sigma). \tag{33}$$

Following eqs. (5) and (6) we can construct the respective PDF

$$Z_{r_\lambda} = \int r_\lambda(\tau) \mathrm{d}\tau, \tag{34}$$

$$\varphi_{r_\lambda}(\sigma) = \frac{r_\lambda(\sigma)}{Z_{r_\lambda}}. \tag{35}$$

Concerning loss weighting, using the weighting function $\lambda$ is equivalent to using the rationale $r_\lambda$, since $\sigma$ and $\epsilon$ are independent we have

$$\mathbb{E}_{\sigma\sim\varphi_r, \varepsilon\sim\mathcal{N}(0,\mathbf{I})} \left[ \lambda(\sigma) || \mathbf{s}_\theta \left( x + \varepsilon\sigma, \sigma \right) - \varepsilon ||^2 \right] = \mathbb{E}_{\varepsilon\sim\mathcal{N}(0,\mathbf{I})} \mathbb{E}_{\sigma\sim\varphi_r} \left[ \lambda(\sigma) || \mathbf{s}_\theta \left( x + \varepsilon\sigma, \sigma \right) - \varepsilon ||^2 \right] \tag{36}$$

$$= \mathbb{E}_{\varepsilon\sim\mathcal{N}(0,\mathbf{I})} \left[ \int || \mathbf{s}_\theta \left( x + \varepsilon\tau, \tau \right) - \varepsilon ||^2 \lambda(\tau) \varphi_r(\tau) d\tau \right] \tag{37}$$

$$\overset{eq.\,(33)}{=} \mathbb{E}_{\varepsilon\sim\mathcal{N}(0,\mathbf{I})} \left[ \int || \mathbf{s}_\theta \left( x + \varepsilon\tau, \tau \right) - \varepsilon ||^2 r_\lambda(\tau) d\tau \right] \tag{38}$$

$$\overset{eq.\,(35)}{=} \mathbb{E}_{\varepsilon\sim\mathcal{N}(0,\mathbf{I})} \left[ Z_{r_\lambda} \int || \mathbf{s}_\theta \left( x + \varepsilon\tau, \tau \right) - \varepsilon ||^2 \varphi_{r_\lambda}(\tau) d\tau \right] \tag{39}$$

$$= Z_{r_\lambda} \mathbb{E}_{\sigma\sim\varphi_{r_\lambda}, \varepsilon\sim\mathcal{N}(0,\mathbf{I})} \left[ || \mathbf{s}_\theta \left( x + \varepsilon\sigma, \sigma \right) - \varepsilon ||^2 \right]. \tag{40}$$

Note that $Z_{r_\lambda}$ is a constant and in consequence we have

$$\min_\theta \mathbb{E}_{\sigma\sim\varphi_r, \varepsilon\sim\mathcal{N}(0,\mathbf{I})} \left[ || \lambda(\sigma) \mathbf{s}_\theta \left( x + \varepsilon\sigma, \sigma \right) - \varepsilon ||^2 \right] = \min_\theta \mathbb{E}_{\sigma\sim\varphi_{r_\lambda}, \varepsilon\sim\mathcal{N}(0,\mathbf{I})} \left[ || \mathbf{s}_\theta \left( x + \varepsilon\sigma, \sigma \right) - \varepsilon ||^2 \right] \tag{41}$$

for any weighting function $\lambda$ and corresponding rationale $r_\lambda$.

## C   RATIONALES

In section 3.3 we touched on some rationales, which we featured in our experiments. Here we provide detailed derivations of these rationales, as well as additional exemplary rationales.

### C.1   SQUARED L2-NORM

Using the proposed framework, we can derive variance schedules directly from metrics, such as the squared L2-norm between densities in the forward diffusion process

$$r_{L2}(\sigma(t)) = \int_{-\infty}^{\infty} |p_{0t} \left( \mathbf{x} | \mathbf{1} \cdot v_{\max} \right) - p_{0t} \left( \mathbf{x} | -\mathbf{1} \cdot v_{\max} \right)|^2 \, \mathrm{d}\mathbf{x}, \tag{42}$$

with respect to variance arising from the upper bound $v_{\max}$ and lower bound $v_{\min} = -v_{\max}$ of any initial value $x(0)$. Commonly, the lower and upper bounds of $x(0)$ are fixed as $-1$ and $1$, respectively. For driftless diffusion processes, we can implement the squared L2-norm as the following

---

**Algorithm 1** Sampling from a discretized probability density function via the generalized inverse.

**Input:** A series of $N$ values $V$ from a rationale $r$, obtained at equidistant steps.

$Z = \sum_{i=1}^{N} V_i$      ▷ normalization to approximate PDF

$F = \left( \frac{1}{Z} V_1, \frac{1}{Z} \sum_{i=1}^{2} V_i, \dots, \frac{1}{Z} \sum_{i=1}^{N} V_i \right)$      ▷ cumulative sum to approximate CDF

$t \sim \mathcal{U}[0,1]$      ▷ sample a percentile

$k = \min \{ i : F_i \geq t \}$      ▷ choose index closest to the percentile (generalized inverse)

**return** $V_k$

---

rationale

$$r_{L2}(\sigma) = \sqrt{2} \left( 1 - \exp\left(-\frac{v_{\max}^2}{\sigma^2}\right) \right), \tag{43}$$

where we abbreviate $\sigma(t)$ with $\sigma$. This rationale favors variances where the densities of maximal and minimal initial values are separable, i.e., variances where details can be restored.

The rationale $r_{L2}$ induces the PDF

$$\varphi_{r_{L2}}(\sigma) = \frac{r_{L2}(\sigma)}{v_{\max}\sqrt{2\pi}}, \tag{44}$$

defined on the domain $\sigma \in (0, \infty)$, and the CDF

$$\Phi_{r_{L2}}(\sigma) = \frac{1}{Z}\sqrt{2} \left( \operatorname{erf}\left(\frac{v_{\max}}{\sigma}\right) v_{\max}\sqrt{\pi} + \sigma \exp\left(-\frac{v_{\max}^2}{\sigma^2}\right) - v_{\max}\sqrt{2} \right), \tag{45}$$

defined on $\sigma \in (0, \infty)$. While the inverse function $\Phi_{r_{L2}}^{-1}$ does not have a trivial closed-form solution, we can use the generalized inverse of the CDF

$$\Phi_{r_{L2}}^{-1}(t) = \inf\left\{\sigma : \Phi_{r_{L2}}(\sigma) \geq t\right\}, \tag{46}$$

and approximate it by discretizing with an appropriate step size, as well as an appropriate $\sigma_{\max}$, see Algorithm 1. While this comes at computational cost, we can compute these values before training or sampling and store them, allowing us to retrieve $\Phi_{r_{L2}}^{-1}(t)$ in constant time. It is important to use a sufficient amount of discretization steps on a restricted domain $[\sigma_{\min}, \sigma_{\max}]$ to approximate a continuous process with respect to $\Phi_{r_{L2}}^{-1}(t)$.

## C.2 GAUSSIAN

Rationales can also follow prior believes, such as assuming that sampling the standard diviation of variance in the forward diffusion process proportional to a normal-distribution $\mathcal{N}\left(0, \sigma_{\mathcal{N}}^2\right)$, centered at 0 with variance $\sigma_{\mathcal{N}}^2$ is beneficial. The rationale of this prior believe can be defined as

$$r_{\mathcal{N}}(\sigma) = \exp\left(-0.5\frac{\sigma^2}{\sigma_{\mathcal{N}}^2}\right), \tag{47}$$

defined on the domain $[0, \infty)$ with the respective PDF

$$\varphi_{r_{\mathcal{N}}}(\sigma) = \frac{\sqrt{2}}{\sigma_{\mathcal{N}}\sqrt{\pi}}r_{\mathcal{N}}(\sigma), \tag{48}$$

and CDF

$$\Phi_{r_{\mathcal{N}}}(\sigma) = \int_0^\sigma \varphi_{r_{\mathcal{N}}}(\tau)\mathrm{d}\tau = \operatorname{erf}\left(\frac{\sigma}{\sqrt{2}\sigma_{\mathcal{N}}}\right), \tag{49}$$

defined on $[0, \infty)$. Here, the CDF does have a closed form solution, which can be used directly as the resulting variance schedule

$$\Phi_{r_{\mathcal{N}}}^{-1}(t) = \operatorname{erf}^{-1}(t)\sqrt{2}\sigma_{\mathcal{N}}. \tag{50}$$

Given that approximately 99.7% of all density lies in the interval $[0, 3\sigma_{\mathcal{N}}]$, this rationale effectively constraints the maximum variance to $3\sigma_{\mathcal{N}}$, oversampling regions near the zero-mean.

## C.3 KL DIVERGENCE

The Kullback-Leibler (KL) divergence of two Gaussians $\mathcal{N}(x; v_{max}, \sigma^2)$ and $\mathcal{N}(x; -v_{max}, \sigma^2)$ is defined by

$$r_D(\sigma) = D\left(\mathcal{N}\left(x; v_{\max}, \sigma^2\right) \,||\, \mathcal{N}\left(x; -v_{\max}, \sigma^2\right)\right) \tag{51}$$

$$= \int \mathcal{N}\left(x; v_{\max}, \sigma^2\right) \log\left(\frac{\mathcal{N}\left(x; v_{\max}, \sigma^2\right)}{\mathcal{N}\left(x; -v_{\max}, \sigma^2\right)}\right) dx \tag{52}$$

$$= \int \mathcal{N}\left(x; v_{\max}, \sigma^2\right) \log\left(\mathcal{N}\left(x; v_{\max}, \sigma^2\right)\right) dx$$
$$- \int \mathcal{N}\left(x; v_{\max}, \sigma^2\right) \log\left(\mathcal{N}\left(x; -v_{\max}, \sigma^2\right)\right) dx \tag{53}$$

$$= \log\left(\frac{\sigma}{\sigma}\right) + \frac{\sigma^2 + (v_{\max} - (-v_{\max}))^2}{2\sigma^2} - \frac{1}{2} \tag{54}$$

$$= \frac{2v_{\max}^2}{\sigma^2}, \tag{55}$$

which on $\sigma \in [\sigma_{\min}, \infty)$ induces the PDF

$$\varphi_{r_D}(\sigma) = \frac{1}{Z_{r_D}} r_D(\sigma), \tag{56}$$

where $Z_{r_D} = \int_{\sigma_{\min}}^{\infty} \frac{2v_{\max}^2}{\sigma^2} d\sigma = \frac{2v_{\max}^2}{\sigma_{\min}}$. The corresponding CDF then is

$$\Phi_{r_D}(\sigma) = \int_{\sigma_{\min}}^{\sigma} \varphi_{r_D}(\tau) d\tau \tag{57}$$

$$= -\frac{2\left(\frac{1}{\sigma} - \frac{1}{\sigma_{\min}}\right) v_{\max}^2}{Z_{r_D}}, \tag{58}$$

with $\sigma \in [\sigma_{\min}, \infty)$. By definition of the normalizing constant $Z_{r_D}$ we have

$$\Phi_{r_D}(\sigma) = -\sigma_{min}\left(\frac{1}{\sigma} - \frac{1}{\sigma_{min}}\right) \tag{59}$$

with the closed-form inverse function

$$\Phi_{r_D}^{-1}(t) = \frac{\sigma_{\min}}{(1-t)}. \tag{60}$$

Remarkably, this removes any necessity for choosing a maximal variance $\sigma_{\max}$.

## C.4 ELUCIDATING THE DESIGN SPACE OF DIFFUSION MODELS

In Karras et al. (2022), no clear variance schedule was defined; rather, a time-discretization effectively served as the variance schedule. The schedule was defined as

$$\Phi_{\text{elucidating}}^{-1}(t) = \left(\sigma_{\max}^{1/\rho} + t\left(\sigma_{\min}^{1/\rho} - \sigma_{\max}^{1/\rho}\right)\right)^\rho, \tag{61}$$

where $\rho$ is a hyper-parameter. Following the relation of the variance schedule and the underlying normalized rationale, we derive

$$\Phi_{\text{elucidating}}(\sigma) = \frac{\sigma^{1/\rho} - \sigma_{\max}^{1/\rho}}{\sigma_{\min}^{1/\rho} - \sigma_{\max}^{1/\rho}}, \tag{62}$$

$$\varphi_{\text{elucidating}}(\sigma) = \frac{\partial}{\partial\sigma}\Phi_{\text{elucidating}}(\sigma) = \frac{\sigma^{(1/\rho)-1}}{\rho\left(\sigma_{\min}^{1/\rho} - \sigma_{\max}^{1/\rho}\right)}, \tag{63}$$

which is the rationale of the time discretization. We also note that the authors apply loss weighting during training, which results in a altered variant of this schedule during training, as proven in appendix B.

Table 3: Results with standard deviations for CIFAR-10 (class-conditional). All models were trained from scratch. The table consists of the best FIDs recorded during training. All FIDs were calculated for five distinct sets of 10K samples and with the CIFAR-10 test-split (10K samples).

| Training \ Sampling | FID $\downarrow$ | | | | |
| --- | --- | --- | --- | --- | --- |
| | **VE** | **L2** | $\mathcal{N}(\mathbf{0}, \mathbf{5^2})$ | **EDM** $(\rho = \mathbf{5})$ | **EDM** $(\rho = \mathbf{7})$ |
| **VE** | 12.28 ±0.16 | 10.65 ±0.09 | **9.17 ±0.09** | 11.94 ±0.13 | 11.90 ±0.08 |
| **L2** | 11.48 ±0.14 | 10.84 ±0.10 | 9.28* ±0.10 | 11.40 ±0.18 | 11.39 ±0.10 |
| $\mathcal{N}(\mathbf{0}, \mathbf{5^2})$ | 12.64 ±0.20 | 11.70 ±0.12 | 10.67 ±0.05 | 11.28 ±0.15 | 11.17 ±0.17 |

Table 4: Results with standard deviations for ImageNet-32 (class-conditional). All models were trained from scratch. The table consists of the best FIDs recorded during training. All FIDs were calculated for four distinct sets of 25K samples and with the ImageNet-32 validation-split (50K samples).

| Training \ Sampling | FID $\downarrow$ | | | | |
| --- | --- | --- | --- | --- | --- |
| | **VE** | **L2** | $\mathcal{N}(\mathbf{0}, \mathbf{5^2})$ | **EDM** $(\rho = \mathbf{5})$ | **EDM** $(\rho = \mathbf{7})$ |
| **VE** | 8.13 ±0.10 | 7.99 ±0.04 | 9.69 ±0.10 | 7.84 ±0.09 | 7.89 ±0.05 |
| **L2** | 6.84 ±0.11 | 6.46 ±0.08 | **5.76 ±0.05** | 6.38 ±0.06 | 6.44 ±0.09 |
| $\mathcal{N}(\mathbf{0}, \mathbf{5^2})$ | 8.29 ±0.02 | 7.84 ±0.05 | 8.64 ±0.05 | 7.06 ±0.06 | 7.01 ±0.10 |

# D    ADDITIONAL EXPERIMENTS

In this section of the appendix, we feature an ablation study w.r.t. the choice of a representative for the $\mathcal{N}\left(0, \sigma_{\mathcal{N}}^2\right)$ rationale, as well as examples of generated images. Further experiments with variance schedules such as the EDM schedule (Karras et al., 2022) with different hyperparameters $\rho$ we also conducted, highlighting the performance of the proposed rationales when sampling.

## D.1    GAUSSIAN RATIONALES

We conducted an ablation study, sampling from a model that has been trained with the VE schedule. Figure 5 shows that the variance schedule of the rationale $\mathcal{N}\left(0, 5^2\right)$ performs best w.r.t. FID.

## D.2    EXEMPLARY IMAGES

In figs. 8 and 9 we display exemplary images for the FID scores reported in tables 1 and 2. All displayed samples were chosen randomly from the generated images, none of them were cherry-picked. We note that all subplots display 100 images, which is a small subset of the 10K generated samples for CIFAR-10 and 50K generated samples for ImageNet-32. The plots are not representative of the generated data as a whole and only serve as visual impressions.

## D.3    CIFAR-10

We conducted additional experiments on CIFAR-10, exploring standard deviations regarding multiple evaluations of FID scores. We focused on the first 400K optimization steps, as models reached their optimal performance within the first 400K steps. All results are listed in table 3 and the FID evolution over optimization steps is shown in fig. 6.

## D.4    IMAGENET-32

We also conducted additional experiments on ImageNet-32, exploring standard deviations regarding multiple evaluations of FID scores. We focused on the entire training. All results are listed in table 4 and the FID evolution over optimization steps is shown in fig. 7.

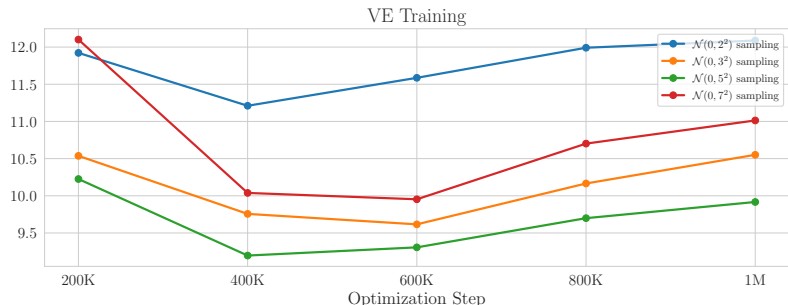

Figure 5: Ablation Study of different $\sigma_{\mathcal{N}}$, sampling from a model that has been trained with the VE schedule.

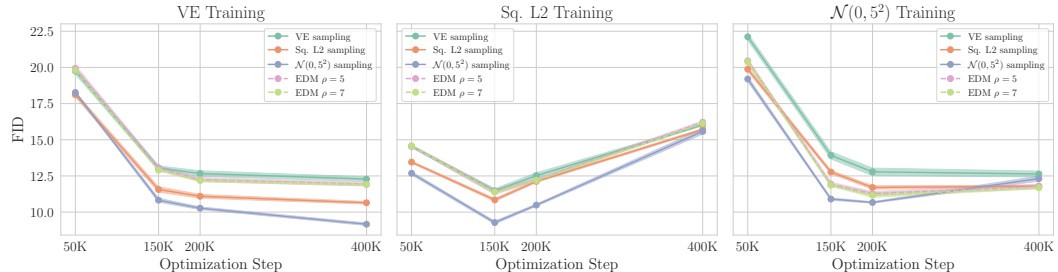

Figure 6: Metrics throughout training (class-conditional) on CIFAR-10 until 400K optimization steps, all metrics are based on 10K samples for all rationales (VE, squared L2-norm, $\mathcal{N}(0, 5^2)$, and EDM). Displayed are FID scores (lower is better) on the CIFAR-10 test-split. All plots have a shared y-axis. The standard deviation of five distinct evaluations of the FID score is underlayed. Each subplot represents using a specific rationale during training with different rationales used during sampling.

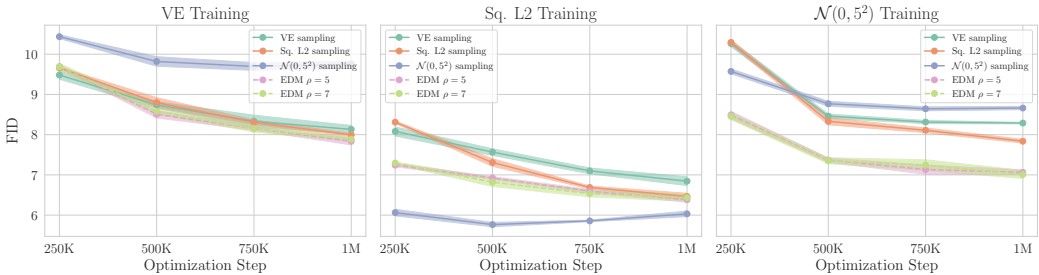

Figure 7: Metrics throughout entire trainings (class-conditional) on ImageNet-32, all metrics are based on 25K generated samples for all rationales (VE, squared L2-norm, $\mathcal{N}(0, 5^2)$, and EDM). Displayed are FID scores (lower is better) on the ImageNet-32 test-split in. All plots have a shared y-axis. The standard deviation of four distinct evaluations of the FID score is underlayed. Each subplot represents using a specific rationale during training with different rationales used during sampling.

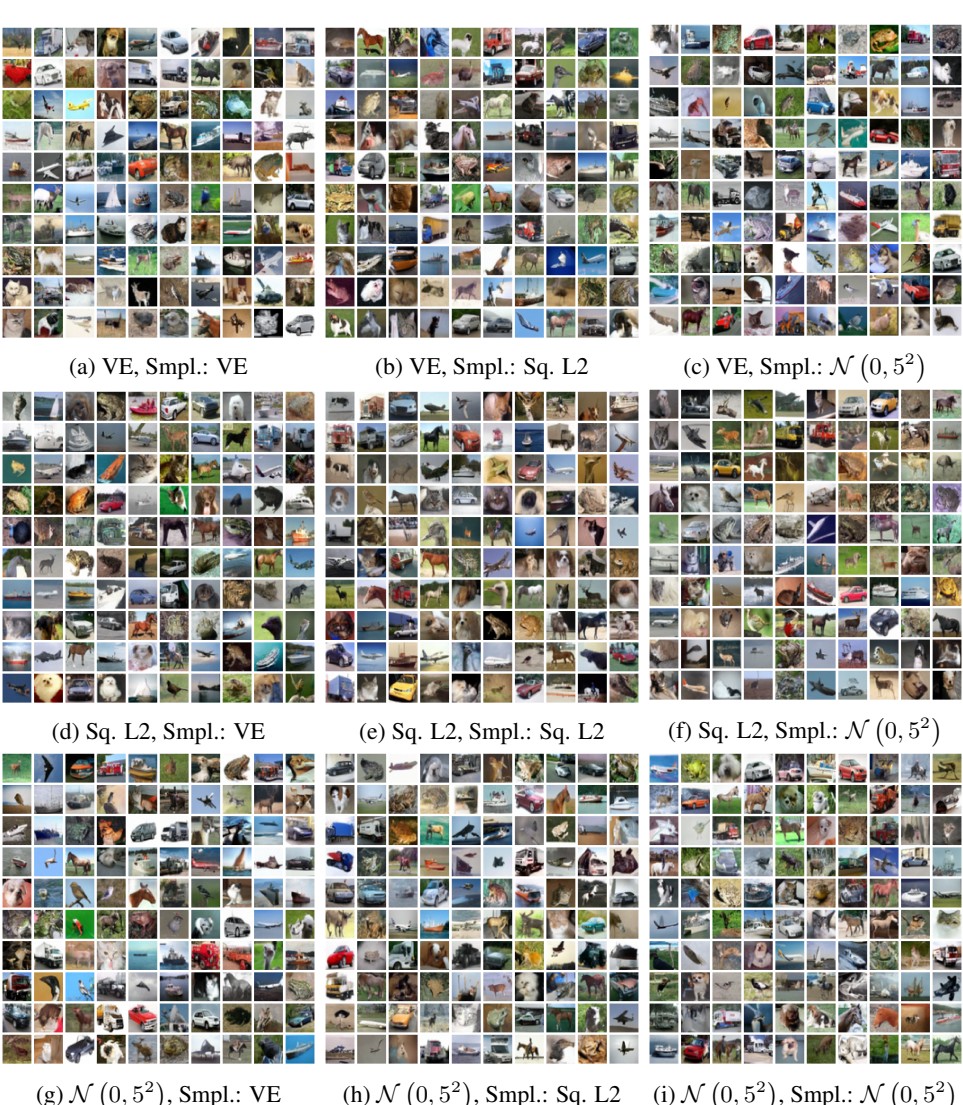

(a) VE, Smpl.: VE      (b) VE, Smpl.: Sq. L2      (c) VE, Smpl.: $\mathcal{N}\left(0, 5^2\right)$

(d) Sq. L2, Smpl.: VE      (e) Sq. L2, Smpl.: Sq. L2      (f) Sq. L2, Smpl.: $\mathcal{N}\left(0, 5^2\right)$

(g) $\mathcal{N}\left(0, 5^2\right)$, Smpl.: VE      (h) $\mathcal{N}\left(0, 5^2\right)$, Smpl.: Sq. L2      (i) $\mathcal{N}\left(0, 5^2\right)$, Smpl.: $\mathcal{N}\left(0, 5^2\right)$

Figure 8: Generated CIFAR-10 Samples that FID scores were reported on. All displayed samples were randomly selected from 10K samples each.

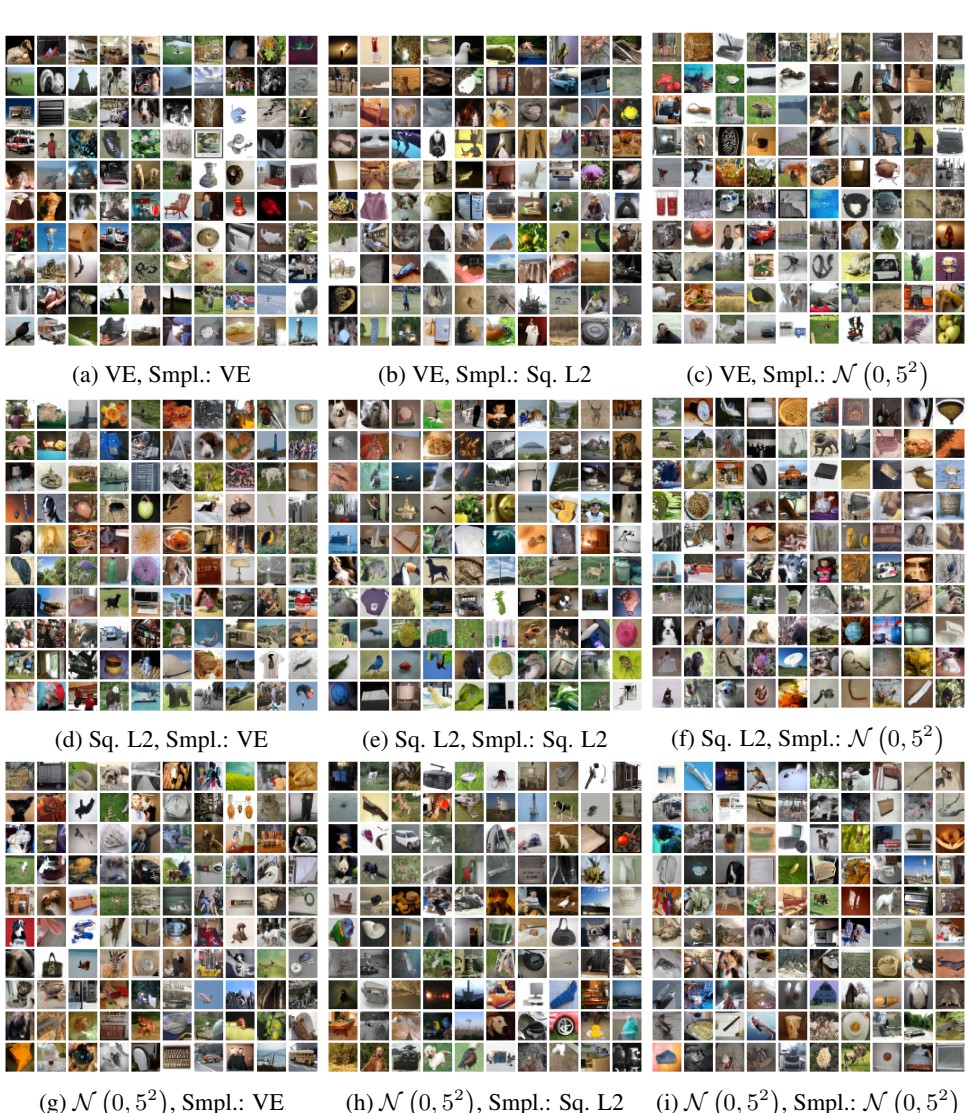

(a) VE, Smpl.: VE     (b) VE, Smpl.: Sq. L2     (c) VE, Smpl.: $\mathcal{N}\left(0, 5^2\right)$

(d) Sq. L2, Smpl.: VE     (e) Sq. L2, Smpl.: Sq. L2     (f) Sq. L2, Smpl.: $\mathcal{N}\left(0, 5^2\right)$

(g) $\mathcal{N}\left(0, 5^2\right)$, Smpl.: VE     (h) $\mathcal{N}\left(0, 5^2\right)$, Smpl.: Sq. L2     (i) $\mathcal{N}\left(0, 5^2\right)$, Smpl.: $\mathcal{N}\left(0, 5^2\right)$

Figure 9: Generated ImageNet-32 Samples that FID scores were reported on. All displayed samples were randomly selected from 50K samples each.

