# OpenReview forum: "Inverse Engineering Diffusion: Deriving Variance Schedules with Rationale"
_ICLR.cc/2025/Conference — Submitted to ICLR 2025_

### Official Review · Reviewer_ZD4R · 2024-10-31

**Soundness:** 2
**Presentation:** 2
**Contribution:** 2
**Rating:** 3
**Confidence:** 3

**Summary:**

The paper proposes to design variance schedules (without drift, identical to noise schedules) for diffusion models in terms of a sampling density of the variance.  This sampling density is derived from the introduced rationale (the unnormalized density) and it is shown how the rationale relates to the variance schedule and2q1` can be used in place of the variance schedule.  Loss weighting is shown to be equivalent to selecting a different rationale at training time than sampling time, and the paper demonstrate that using a different schedule at sampling than training time can give improved performance.  The paper analyzes multiple metrics (FID, Inception Score) across two datasets (CIFAR-10, ImageNet-32), including metric measurements as a function of optimization step.

**Strengths:**

- Provides empirical evidence that separating out the choice of noise schedule at training and sampling time can be useful
- Reasonably large effects on sampling performance are shown as a function of the noise schedule, reconfirming that the choice of noise schedule can substantively alter training and sampling
- Demonstrates rationales without a closed form solution for sigma(t) may still be used, albeit with additional computational expense

**Weaknesses:**

- The central concept of the paper, the rationale, is the unnormalized density of the variances.  It's introduced and described in an unclear fashion, yet the resulting computations follow from the 1-1 invertible relationship between sigma(t) and time.  Focusing design on the unnormalized density rather than the sigma(t) function is reasonable, especially given the connection to loss weighting as described in Section 3.4, but the reparameterization is straightforward.

- Many of the theoretical contributions have been used previously.  Including, for example, the ability to use different noise schedules at sampling and training, that loss can be written entirely in terms of SNR sampling (or equivalently variance sampling without mean scaling), and that loss weighting is equivalent to sampling from a different distribution.  The paper could be improved by clarifying what is novel.

- The empirical results do not appear to lead to strong conclusions about what noise schedule should be used at training or sampling time.  While the conclusion that separating training and sampling schedules is reasonable and interesting, it follows from the pre-existing prevalence of loss weighting schemes.  Further, the variances schedules examined (VE, L2-Norm, and Gaussian) only include one from prior literature which limits the significance of the comparison.  The introduced L2-Norm and Gaussian rationales receive relatively little motivation.

**Questions:**

- I appreciate that the authors included the standard deviation for IS in their figures and tables.  Given the displayed error bars, are the differences between variance schedules statistically significant? In Table 1 VE, sampling and L2 training is bolded, yet the +- error overlaps quite strongly with other elements in the table.

---

> ### Author Response · Authors · 2024-11-21
> **Rebuttal by Authors**
>
> We sincerely thank the reviewer for their thoughtful feedback, which highlights areas for improvement while acknowledging the strengths of our work. Below, we provide detailed responses to the reviewers comments and questions:
>
> **(i) Weakness 2:**
> - Our main contribution is the unification of variance scheduling, variance distribution, and loss weighting into a single framework via the concept of rationales. This eliminates the need for loss weighting, as demonstrated in Equations (22-24), and offers theoretical insights into curated loss weighting schemes from prior works [1, 2, 3].
> - While these individual steps may appear intuitive, the comprehensive integration of these elements into a rationale-based framework is, to the best of our knowledge, novel and has not been previously explored.
> - We also want to state that the generalized inverse of the CDF can be used to define variance schedules that would have been unattainable through conventional methods. To the best of our knowledge, this is a novelty in the context of diffusion models. This allows us to explore rationales such as the squared L2-norm, which yields faster convergence across both datasets.
>
> We have made changes to the introductory section of the paper to better clarify these novelties, see changes on **lines 065-067, page 2**.
>
> [1] Karras et al. "Elucidating the Design Space of Diffusion-Based Generative Models", NeurIPS, 2022.
> [2] Song, Yang et al. "Consistency Models" ICML, 2023.
> [3] Song, Yang et al. "Improved Techniques for Training Consistency Models" ICLR, 2023.
>
> **(ii) Weakness 3 and Question 1:**
>
> - To address the reviewers request to consider more schedules from prior literature, we expanded our evaluation to EDM schedules from Karras et al. [1] (see also **Appendix C.4** for respective rationale) with two common hyperparameter-selections $\rho=5$ and $\rho=7$. In the following table, we display the results. For detailed plots, please consider **Figures 6 & 7** in the updated document.
>
> **CIFAR-10:**
>
> | Training\Sampling        | **VE**                  | **L2**                  | **N(0, 5²)**             | **EDM (ρ=5)**           | **EDM (ρ=7)**           |
> |--------------------------|-------------------------|-------------------------|--------------------------|-------------------------|-------------------------|
> | **VE**                  | 12.28 ± 0.16           | 10.65 ± 0.09           | **9.17 ± 0.09**          | 11.94 ± 0.13           | 11.90 ± 0.08           |
> | **L2**                  | 11.48 ± 0.14           | 10.84 ± 0.10           | 9.28* ± 0.10          | 11.40 ± 0.18           | 11.39 ± 0.10           |
> | **N(0, 5²)**           | 12.64 ± 0.20           | 11.70 ± 0.12           | 10.67 ± 0.05            | 11.28 ± 0.15           | 11.17 ± 0.17           |
>
>
> **ImageNet-32:**
>
> | Training\Sampling        | **VE**              | **L2**              | **N(0, 5²)**         | **EDM (ρ=5)**       | **EDM (ρ=7)**       |
> |--------------------------|---------------------|---------------------|----------------------|---------------------|---------------------|
> | **VE**                  | 8.13 ± 0.10        | 7.99 ± 0.04        | 9.69 ± 0.10         | 7.84 ± 0.09        | 7.89 ± 0.05        |
> | **L2**                  | 6.84 ± 0.11        | 6.46 ± 0.08        | **5.76 ± 0.05**     | 6.38 ± 0.06        | 6.44 ± 0.09        |
> | **N(0, 5²)**           | 8.29 ± 0.02        | 7.84 ± 0.05        | 8.64 ± 0.05         | 7.06 ± 0.06        | 7.01 ± 0.10        |
>
>
> Results confirmed the superior sampling properties of our proposed rationales. These results are now included in Appendix D.3 and D.4 (see pages 17 & 18).
> - We agree that the differences in IS scores on CIFAR-10 are within the margins of their respective standard deviations. However, on ImageNet-32, the improvements are statistically robust. For clarity, we bolded the highest mean values, and results with overlapping standard deviations were marked explicitly with an asterisk in each table. We are open to suggestions for alternative highlighting methods.
> - We improved the motivation in sections 3.3.2 and 3.3.3 (pages 5 & 6) respectively, highlighting why these two rationales are of particular interest to driftless diffusion processes:
>
> *The squared L2-norm rationale is of particular interest for driftless processes. With this approach, we can ascribe high density to points in time of the diffusion process where distributions of different originating means (i.e., $x(0)$) should yield high separability. Oversampling these regions could lead to more detailed samples.*
>
> *The Gaussian rationale is of particular interest for a parameterized variance schedule that can be tuned to different data-domains. The normal distribution is commonly used to model probabilistic processes due to its good generalization properties in most use-cases.*

---

> ### Author Response · Authors · 2024-11-21
>
> **(iii) Weakness 1:**
> - We thank the reviewer for acknowledging the practicality of our focus on unnormalized density-based variance scheduling. We aimed to thoroughly introduce the concept of the rationale in section 3.1, featuring all steps from the rationale to its respective PDF & CDF and the respective variance schedule (see also **Figure 1** for an overview of the relations between steps). We also provided examples for the derivation of variance schedules from rationales in 3.3.2, 3.3.3 and in Appendix C. If any aspects of the rationale’s presentation remain unclear, we would appreciate additional clarification to address these points more effectively.
>
> Lastly, we would like to point out that the computational expense of our method is negligible. The required values can be precomputed, and sampling during training can be efficiently implemented using optimized algorithms, such as `np.random.choice`. During inference, sigma values are precomputed once and reused, resulting in no significant computational cost.
>
> We hope these clarifications sufficiently address the reviewer's concerns and highlight the robustness and novelty of our work. We sincerely thank the reviewer again for the valuable feedback and kindly ask the reviewer to reconsider their initial assessment.

---

> > ### Comment · Reviewer_ZD4R · 2024-11-22
> >
> > I thank the authors for their response.  While weakness 3 has been addressed, weakness 1 (unclear conceptual necessity of the framework) and 2 (limited theoretical contributions) are not.  In my view, the empirical results are not strong enough to overcome these weaknesses.

---

> > > ### Author Response · Authors · 2024-11-24
> > >
> > > We appreciate the reviewer's reply and the reviewer's acknowledgment that weakness 3 has been fully addressed.
> > >
> > > We elaborated extensively on (1) the conceptual necessity of the framework and (2) the theoretical contribution. We highlighted that through our theoretical contributions novel variance schedules such as the variance schedule following the squared L2-Norm rationale can be explored. This would not have been possible without the proposed framework and necessitates our work, especially in light of the performance of the squared L2-Norm rationale.
> > >
> > > We highlighted that the squared L2-Norm rationale leads to faster convergence across all experiments and datasets when used in training. This is emphasized by the FID scores, and the standard deviation of Inception Scores on ImageNet-32, where all results within margin of the best score were obtained from the squared L2-Norm rationale (including the best result itself), as well as the additional experiments, including extended prior literature, which we added in the rebuttal (see Tables **1,2,3, & 4**) and **Figures 3,4, 6 &7**.
> > >
> > > We believe that the exploration of variance schedules, which would have been unattainable through conventional methods, and the strong empirical results thereof on ImageNet-32 and CIFAR-10  justify the proposed concept of the rationale.
> > >
> > > We hope these clarifications sufficiently address the reviewer's concerns. We sincerely thank the reviewer again for the valuable feedback and kindly ask the reviewer to reconsider their initial assessment.

---

### Official Review · Reviewer_oBtB · 2024-11-04

**Soundness:** 3
**Presentation:** 3
**Contribution:** 3
**Rating:** 5
**Confidence:** 3

**Summary:**

This study introduces a new framework for understanding and designing variance schedules in score-based diffusion models. It reinterprets variance schedules as being derived from probabilistic rationales, where the variance distribution is framed as a CDF with an associated PDF.
Viewing variance schedules through this probabilistic lens, the approach enables direct engineering of schedules based on specific rationales, offering flexibility beyond traditional methods.
This design allows for schedules with varied dynamics, such as rapid convergence to zero variance or extended periods in high-variance regions, making it possible to create novel schedules that conventional techniques cannot achieve.
The framework is showcased using a process "inverse engineering" diffusion by constructing these schedules from the inverse of the CDF, providing a new means of controlling the diffusion process based on targeted variance rationales.
The method is validated using empirical evaluations on benchmark datasets like CIFAR-10, results show that variance schedules based on different rationales can improve image generation quality, evidenced by superior FID scores.

**Strengths:**

The manuscript is clear and well-organized. This framework offers a unified perspective that clarifies the relationship between variance schedules and probabilistic rationales, revealing new opportunities to enhance performance in diffusion model applications. The approach is innovative, deriving variance schedules from probabilistic rationales, and is empirically validated on ImageNet-32, effectively expanding the design space for diffusion models.

**Weaknesses:**

- Minor typo on line 52 r/weighs/weights
- Insufficient guidance on choosing the most effective rationale for specific tasks or datasets
- A sensitivity analysis showing how different parameter values affect model performance or convergence would provide actionable insights for tuning the method in diverse settings.

**Questions:**

- Refer to the issues noted in the weaknesses section.
- How do computational requirements vary across different rationale-based schedules?
- Is there a recommended method for tuning parameters within each selected rationale?

---

> ### Author Response · Authors · 2024-11-21
> **Rebuttal by Authors**
>
> We sincerely thank the reviewer for their thoughtful feedback, which highlights the novelty and unifying perspective our work. The reviewer raises questions regarding the sensitivity of hyperparamters in rationales. Below, we provide detailed responses to the reviewers comments and questions:
>
> **(i) Weakness 1 and Question 1:**
> - We appreciate the reviewer highlighting the typos, which have been corrected and we thank the reviewer for helping improve the clarity of our work.
>
> **(ii) Weakness 2, 3 and Question 1:**
> - Parameter tuning for the normal distribution rationale was performed through sensitivity analysis regarding the hyper-parameter $\sigma_{\mathcal{N}}$ (see Appendix  D.1), where we found an optimal $\sigma_\mathcal{N} = 5$ on CIFAR-10. For consistency, $\sigma_\text{max}$ was set to $3\sigma_\mathcal{N}$, encompassing 99.73% of the density.
> - The squared L2-Norm rationale has minimal parameterization, apart from $\sigma_\text{max}$, which we matched to the VE rationale for direct comparison. As shown in **Figures (3,4,6,7)**, despite similar density patterns (see **Figure 2**), training with the squared L2-Norm rationale demonstrated superior convergence on ImageNet in particular, showcasing its practical effectiveness. The parameterization of the VE schedule follows the prior work of Song et al. [1], which achieved the best overall FID results in their experiments.
>
> [1] Song, Yang et al. "Score-Based Generative Modeling through Stochastic Differential Equations" ICLR, 2021.
>
> **(iii) Question 2:**
> - Our approach does not introduce significant computational overhead. Equation (9) provides the general method for sampling variance distributions, in practice, we can rely on efficient implementations such as `np.random.choice`, which samples from discrete probabilities (normalized discretization within the considered interval) in negligible time. Training-time complexity remains unaffected, as sampling during training is computationally negligible relative to the overall training time. For inference, precomputed sigma values eliminate additional costs entirely.
>
> **(iv) Question 3:**
> - We recommend a grid-search search for hyperparameter tuning of rationales that rely on hyperparameters, as demonstrated in Appendix D.1. This ensures that model performance can be optimized across diverse settings.
>
> We hope these clarifications sufficiently address the reviewer's concerns about our work. We sincerely thank the reviewer again for the valuable feedback and kindly ask the reviewer to reconsider their initial assessment.

---

> > ### Author Response · Authors · 2024-11-25
> >
> > As the discussion period ends on November 26, we want to ensure that all your questions have been thoroughly addressed by our rebuttal.
> >
> > Your feedback is instrumental to us, and we would be grateful if you could spare a moment to provide a final rating and share your thoughts on our rebuttal.

---

### Official Review · Reviewer_jqJQ · 2024-11-06

**Soundness:** 2
**Presentation:** 2
**Contribution:** 2
**Rating:** 5
**Confidence:** 2

**Summary:**

This work introduces an approach to understand and design variance schedules in diffusion models, particularly in the context of score-based diffusion frameworks. It emphasizes the importance of the variance schedule in both training and sampling phases, proposing a probabilistic rationale for constructing these schedules. The paper introduces the concept of "rationales," which are functions that define how variance is distributed throughout the diffusion process. By treating the inverse of a variance schedule as a cumulative distribution function (CDF) and its derivative as a probability density function (PDF), the framework allows for more flexible and diverse variance schedules. The paper demonstrates that different schedules—such as those derived from the squared L2-norm or the VE schedule—can impact both model convergence during training and performance during sampling.

**Strengths:**

1. The introduction of the "rationale" concept offers a new way to understand and design variance schedules, allowing for better control over the behavior of the diffusion process.
2. The framework leads to noticeable improvements in image generation, as demonstrated by the enhanced performance metrics (e.g., FID) across different datasets like CIFAR-10 and ImageNet-32.
3. The comprehensive empirical evaluation provides valuable insights into the impact of different variance schedules on model convergence and sample quality.

**Weaknesses:**

1. The proposed framework introduces complexity by requiring a detailed understanding of probability distributions (CDFs, PDFs) and variance schedules, which may be difficult for some practitioners and impact its applicability. While the rationale approach is novel, it may be challenging for users to easily select or engineer appropriate rationales in real-world applications, especially without closed-form solutions for some CDFs.
2. In some cases, such as training with the squared L2-norm rationale, the model showed overfitting on smaller datasets (e.g., CIFAR-10), which could be problematic for practical use in certain domains or tasks.
3. Some of results shown in section 4 results seem not statistical significantly different if we consider its variance, for example in table 1 and table 2. Also in Figure 2 (b), the 3rd plot, there is extremely small difference among V2,  squared L2 sampling, normal sampling; and their superiority order change as number of optimization step increase.
4. some typos: row 132, "defintion"; row 478 has " 1×, 2×, 2×, and 2× the base feature-size": not sure if this is expected.

**Questions:**

1. What are the computational costs associated with using more complex rationales that do not have closed-form inverse solutions? How does this impact scalability for large-scale training or real-time inference?
2. Can the performance improvements in FID be sustained across a wider range of datasets, particularly those that are more challenging or less well-suited to the proposed variance schedules?

---

> ### Author Response · Authors · 2024-11-21
> **Rebuttal by Authors**
>
> We sincerely thank the reviewer for their thoughtful feedback, which focuses on the statistical significance of the experiments and computational cost. Below, we provide detailed responses to the reviewers comments and questions:
>
> **(i) Weakness 1 and Question 1:**
> - Our framework requires only a probability density function and its corresponding cumulative distribution function for variance scheduling, which can be implemented straightforward using Equations (5) and (6). The rationale-based approach is designed to be intuitive while providing greater control over the resulting variance schedule.
> - Even when the inverse of Equation (6) is not available in closed-form, equation (9) ensures efficient approximation of the true inverse function via discretization. Sampling during training can be implemented with highly optimized algorithms, such as `np.random.choice`, which evaluates sampling in a negligible amount of time relative to the GPU-days needed for training. For inference, sigma values are pre-computed once and reused, incurring no significant computational cost.
>
> **(ii) Weakness 2:**
> - Overfitting is a common challenge in generative models. A faster-converging training, achieved via the proposed squared L2-Norm rationale, may lead to earlier overfitting. However, this behavior is offset by its efficiency in achieving high-quality results within fewer optimization steps.
> - We emphasize that the squared L2-Norm rationale performed consistently well across datasets and should be regarded as an efficient option rather than a limitation, delivering higher image quality within fewer optimization steps.
>
> **(iii) Weakness 3 and Question 2:**
> - Due to time constraints and the overall computational cost of training diffusion models, we restricted our experiments to CIFAR-10 and ImageNet-32, which are standard benchmarks in diffusion models. These datasets capture a range of complexities and serve as strong baselines.
> - To address the reviewers concern about statistical significance, we conducted additional experiments with repeated, distinct samplings and report the average FID, as well as standard deviations of FID scores. These results (see updated **Figures 6 & 7** and **Tables 3 & 4** in Appendix D.3 & D.4) confirm the statistical robustness of our findings w.r.t. FID. We also extended the evaluation of prior work to feature samplings with EDM schedules [1], highlighting the efficacy of our methods.
>
> **CIFAR-10:**
>
> | Training\Sampling        | **VE**                  | **L2**                  | **N(0, 5²)**             | **EDM (ρ=5)**           | **EDM (ρ=7)**           |
> |--------------------------|-------------------------|-------------------------|--------------------------|-------------------------|-------------------------|
> | **VE**                  | 12.28 ± 0.16           | 10.65 ± 0.09           | **9.17 ± 0.09**          | 11.94 ± 0.13           | 11.90 ± 0.08           |
> | **L2**                  | 11.48 ± 0.14           | 10.84 ± 0.10           | 9.28* ± 0.10          | 11.40 ± 0.18           | 11.39 ± 0.10           |
> | **N(0, 5²)**           | 12.64 ± 0.20           | 11.70 ± 0.12           | 10.67 ± 0.05            | 11.28 ± 0.15           | 11.17 ± 0.17           |
>
>
> **ImageNet-32:**
>
> | Training\Sampling        | **VE**              | **L2**              | **N(0, 5²)**         | **EDM (ρ=5)**       | **EDM (ρ=7)**       |
> |--------------------------|---------------------|---------------------|----------------------|---------------------|---------------------|
> | **VE**                  | 8.13 ± 0.10        | 7.99 ± 0.04        | 9.69 ± 0.10         | 7.84 ± 0.09        | 7.89 ± 0.05        |
> | **L2**                  | 6.84 ± 0.11        | 6.46 ± 0.08        | **5.76 ± 0.05**     | 6.38 ± 0.06        | 6.44 ± 0.09        |
> | **N(0, 5²)**           | 8.29 ± 0.02        | 7.84 ± 0.05        | 8.64 ± 0.05         | 7.06 ± 0.06        | 7.01 ± 0.10        |
>
> We further changed the presentation in all tables to improve the interpretability of results (see also **Tables 1 & 2**). Results with overlapping standard deviations were marked explicitly with an asterisk. Hightlithing these overlaps also emphasizes that the squared L2-Norm rationale consistently led to faster convergence for ImageNet-32 in particular, as the only Inception Scores with overlaps originated from using the squared L2-norm rationale during ImageNet training (**see Table 2**).
>
> [1] Karras et al. "Elucidating the Design Space of Diffusion-Based Generative Models", NeurIPS, 2022.
>
> **(iv) Weakness 4:**
> - We thank the reviewer for noting the typos. We have corrected the errors on lines 132 and 478, along with other minor issues throughout the text to improve clarity and presentation.

---

> ### Author Response · Authors · 2024-11-21
>
> We hope these clarifications sufficiently address the reviewer's concerns about our work. We sincerely thank the reviewer again for the valuable feedback and kindly ask the reviewer to reconsider their initial assessment.

---

> > ### Author Response · Authors · 2024-11-25
> >
> > As the discussion period ends on November 26, we want to ensure that all your questions have been thoroughly addressed by our rebuttal.
> >
> > Your feedback is instrumental to us, and we would be grateful if you could spare a moment to provide a final rating and share your thoughts on our rebuttal.

---

### Author Response · Authors · 2024-11-21
**Overall Response**

Dear Reviewers and Respected Area Chair,

We sincerely thank the reviewers for their insightful feedback and constructive comments, which have been instrumental in refining our work. We also appreciate the recognition of several strengths, including:

- The introduction of the *rationale* concept, which provides a novel framework for understanding and designing variance schedules. This approach offers enhanced control over the diffusion process, leading to improved image generation quality, as demonstrated by lower FID scores on CIFAR-10 and ImageNet-32.

- The innovative derivation of variance schedules based on probabilistic rationales, which broadens the design space for diffusion models.

- The acknowledgment that rationales without closed-form solutions remain usable, leading to the exploration of novel variance schedules that are unattainable through conventional methods

We understand the concern about computational complexity and clarify that our approach does not introduce significant additional computation due to pre-computation during sampling and the efficient implementation of `np.random.choice`, which is used during training.

We made changes to the paper on pages 2,6,8,9 and in the appendix on pages 17 & 18. All changes to the initial submission have been highlighted in orange. We want to highlight **Figures 6 & 7** and **Tables 3 & 4** in particular, featuring additional evaluations of our methods, and also including more methods from prior literature.

Detailed responses to individual reviewer comments are provided below. Thank you again for your thoughtful suggestions and insights. We welcome any further questions or feedback.

---

### Meta-Review · Area_Chair_pqRs · 2024-12-20

**Metareview:**

The paper introduces a novel framework for designing variance schedules in diffusion models by interpreting the inverse of the variance schedule as a cumulative distribution function. This approach allows for the direct engineering of schedules based on probabilistic rationales. The work demonstrates improved performance in generative tasks and highlights the effectiveness of decoupling training and sampling schedules.

The reviewers pointed out obstacles to practical use and inadequate experimental performance, as well as novelty and theoretical contributions. Some of the points were addressed by rebuttals, but not enough to overturn other important points.

**Additional Comments On Reviewer Discussion:**

jqJQ and oBtB pointed out obstacles to practical use and inadequate experimental performance; ZD4R added that the approach was too natural and lacked novelty and theoretical contributions.

---

### Decision · Program_Chairs · 2025-01-22

Reject